# Comment on "Synergetic use of IASI and TROPOMI space borne sensors for generating a tropospheric methane profile product"

Simone Ceccherini

Istituto di Fisica Applicata "Nello Carrara" del Consiglio Nazionale delle Ricerche, Via Madonna del Piano 10, 50019 Sesto Fiorentino, Italy

*Correspondence to*: Simone Ceccherini (S.Ceccherini@ifac.cnr.it)

**Abstract.** A great interest is growing about methods that combine measurements from two or more instruments that observe the same species either in different spectral regions or with different geometries. Recently, a method based on the Kalman filter has been proposed to combine IASI (Infrared Atmospheric Sounding Interferometer) and TROPOMI (TROPOspheric Monitoring Instrument) methane products. We show that this method is equivalent to the Complete Data Fusion method. Therefore, the choice between these two methods is driven only by the advantages of the different implementations. From the comparison of the two methods a generalization of the Complete Data Fusion formula, which is valid also in the case that the noise error covariance matrices of the fused products are singular, is derived.

## 1 Introduction

Remote atmospheric measurements are obtained with retrieval techniques (Rodgers, 2000) that determine the distribution of an atmospheric parameter by fitting forward model simulations to the available observations. Generally, the fitting procedure consists in the minimization of a cost function made of the summation of two terms. The first term is the square of the differences between observations and simulations, weighted with the inverse covariance matrix (CM) of the observations and the second term is a constraint that reflects our a priori knowledge of the distribution of the atmospheric parameter. When two or more instruments sound the same portion of atmosphere and observe the same atmospheric parameter either in different spectral regions or with different geometries, the problem arises to identify the best strategy to combine the different measurements in order to exploit all the information that is provided by the observations in presence of the retrieval constraints.

Recently, Schneider et al. (2021) presented a method for the combined use of the individual retrieval products of IASI (Infrared Atmospheric Sounding Interferometer) and TROPOMI (TROPOspheric Monitoring Instrument). The method is based on the Kalman filter (Rodgers, 2000) and, as stated by the authors, is largely equivalent to using the spectra of the different sensors together in a single retrieval procedure (simultaneous retrieval). A method based on the same principle, i.e. the application of the Kalman filter for combining different satellite sensor observations, was already presented in Warner et al. (2014) and used to combine CO products of AIRS (Atmospheric InfraRed Sounder) and TES (Tropospheric Emission Spectrometer) as well of AIRS and MLS (Microwave Limb Sounder).

The purpose of this comment is to study the relationship between the method presented in Schneider et al. (2021) and the Complete Data Fusion (CDF) method presented in Ceccherini et al. (2015). CDF too uses the output of the individual retrievals and, in the case that the linear approximation holds in the range of the retrieved products, provides products equivalent to those of the simultaneous retrieval.

The formulas of the two methods are algebraically different but the connection of both methods with the simultaneous retrieval suggests a close relationship between them. It is important to understand whether the two methods are equivalent (that is they provide the same results) and otherwise to identify the reasons of the differences, because this analysis allows to highlight the advantages of using one method or the other. In effect, in Sections 2 we prove the equivalence between the two methods and in Section 3 we draw the conclusions.

## 2 Equivalence between data fusion with Kalman filter and CDF

In this Section, we prove the equivalence of the method proposed in Schneider et al. (2021) and the CDF method proposed in Ceccherini et al. (2015).

We start from Eq. (A9) of Schneider et al. (2021):

$$\hat{\mathbf{x}}^a = \hat{\mathbf{x}}^b + \mathbf{M}\left[\hat{\mathbf{x}}^o - \mathbf{H}\hat{\mathbf{x}}^b\right], \tag{1}$$

where $\hat{\mathbf{x}}^a$ is the analysis state, $\hat{\mathbf{x}}^b$ is the background state, $\hat{\mathbf{x}}^o$ is the new observation, $\mathbf{M}$ is the Kalman gain matrix and $\mathbf{H}$ is the measurement forward operator. In our case, this equation determines the product of the fusion of two measurements (the analysis state) by combining the first measurement (the background state) with the information provided by the second measurement (the new observation).

We substitute in Eq. (1) the different quantities as given in Appendix A2.3 of Schneider et al. (2021). From Eq. (A14), (A15) and (A13) we have, respectively:

$$\hat{\mathbf{x}}^b = \hat{\mathbf{x}}_1 - \mathbf{x}_a, \tag{2}$$

$$\hat{\mathbf{x}}^o = \hat{\mathbf{x}}_2 - \mathbf{x}_a \tag{3}$$

and

$$\mathbf{H} = \mathbf{A}_2, \tag{4}$$

where $\hat{\mathbf{x}}_1$ and $\hat{\mathbf{x}}_2$ are the two individual retrieval products, $\mathbf{x}_a$ is the a priori state vector and $\mathbf{A}_2$ is the averaging kernel matrix of $\hat{\mathbf{x}}_2$. The Kalman gain matrix $\mathbf{M}$ is given by Eq. (A17) that using Eq. (A4) becomes:

$$\mathbf{M} = \left(\mathbf{F}_1 + \mathbf{F}_2 + \mathbf{S}_a^{-1}\right)^{-1}\left(\mathbf{F}_2 + \mathbf{S}_a^{-1}\right), \tag{5}$$

where $\mathbf{S}_a$ is the a priori CM and we have introduced the Fisher information matrices (Fisher, 1935; Ceccherini et al., 2012) $\mathbf{F}_1$ and $\mathbf{F}_2$, which are given by:

$$\mathbf{F}_i = \mathbf{K}_i^T \mathbf{S}_{\mathbf{y}_i,\mathrm{n}}^{-1} \mathbf{K}_i = \mathbf{A}_i^T \mathbf{S}_{\hat{\mathbf{x}}_i,\mathrm{n}}^{-1} \mathbf{A}_i = \mathbf{S}_{\hat{\mathbf{x}}_i}^{-1} \mathbf{A}_i \qquad i = 1, 2, \tag{6}$$

$\mathbf{K}_i$ being the Jacobian matrices of the forward models, $\mathbf{S}_{\mathbf{y}_i,\mathrm{n}}$ the CMs for noise on the measured radiances $\mathbf{y}_i$, $\mathbf{S}_{\hat{\mathbf{x}}_i,\mathrm{n}}$ the noise error CMs of the retrieved states and $\mathbf{S}_{\hat{\mathbf{x}}_i}$ the a posteriori CMs. The Fisher information matrices are useful quantities that fully describe the information provided by the observations on the state vectors and are easily obtained using either the quantities that characterise the observations or those that characterise the measurements. In the case that the matrices $\mathbf{S}_{\hat{\mathbf{x}}_i,\mathrm{n}}$ are singular, Eq. (6) is still valid replacing the matrices $\mathbf{S}_{\hat{\mathbf{x}}_i,\mathrm{n}}^{-1}$ with the generalized inverse matrices (Kalman, 1976) $\mathbf{S}_{\hat{\mathbf{x}}_i,\mathrm{n}}^{\#}$, as shown in the appendix of Ceccherini et al. (2012). Eqs. (5, 6) show that the Kalman gain matrix takes into account the uncertainties of both measurements as well of the a priori state vector when we combine them using Eq. (1).

In analogy with Eqs. (2) and (3), we introduce the quantity $\hat{\mathbf{x}}_f$ such as the analysis state $\hat{\mathbf{x}}^a$ is the difference between $\hat{\mathbf{x}}_f$ and $\mathbf{x}_a$:

$$\hat{\mathbf{x}}^a = \hat{\mathbf{x}}_f - \mathbf{x}_a. \tag{7}$$

Substituting Eqs. (2-5) and (7) in Eq. (1), we obtain:

$$\hat{\mathbf{x}}_f - \mathbf{x}_a = \hat{\mathbf{x}}_1 - \mathbf{x}_a + \left(\mathbf{F}_1 + \mathbf{F}_2 + \mathbf{S}_a^{-1}\right)^{-1}\left(\mathbf{F}_2 + \mathbf{S}_a^{-1}\right)\left[\hat{\mathbf{x}}_2 - \mathbf{x}_a - \mathbf{A}_2\left(\hat{\mathbf{x}}_1 - \mathbf{x}_a\right)\right]. \tag{8}$$

Using Eqs. (A2-A3) of Schneider et al. (2021) and Eq. (6), we have:

$$\mathbf{A}_i = \left(\mathbf{F}_i + \mathbf{S}_a^{-1}\right)^{-1}\mathbf{F}_i \qquad i = 1, 2, \tag{9}$$

therefore, Eq. (8) can be written as:

$$\hat{\mathbf{x}}_f - \mathbf{x}_a = \left[ \mathbf{I} - \left( \mathbf{F}_1 + \mathbf{F}_2 + \mathbf{S}_a^{-1} \right)^{-1} \mathbf{F}_2 \right] \left( \hat{\mathbf{x}}_1 - \mathbf{x}_a \right) + \left( \mathbf{F}_1 + \mathbf{F}_2 + \mathbf{S}_a^{-1} \right)^{-1} \left( \mathbf{F}_2 + \mathbf{S}_a^{-1} \right) \left( \hat{\mathbf{x}}_2 - \mathbf{x}_a \right) =$$
$$= \left( \mathbf{F}_1 + \mathbf{F}_2 + \mathbf{S}_a^{-1} \right)^{-1} \left[ \left( \mathbf{F}_1 + \mathbf{S}_a^{-1} \right) \left( \hat{\mathbf{x}}_1 - \mathbf{x}_a \right) + \left( \mathbf{F}_2 + \mathbf{S}_a^{-1} \right) \left( \hat{\mathbf{x}}_2 - \mathbf{x}_a \right) \right] \qquad (10)$$

Consistently with Eq. (3) of Ceccherini et al. (2015), we introduce the $\boldsymbol{\alpha}_i$ quantities, such as:

$$\hat{\mathbf{x}}_i - \mathbf{x}_a = \boldsymbol{\alpha}_i - \mathbf{A}_i \mathbf{x}_a \qquad\qquad i = 1, 2 \qquad (11)$$

and Eq. (10) becomes:

$$\hat{\mathbf{x}}_f - \mathbf{x}_a = \left( \mathbf{F}_1 + \mathbf{F}_2 + \mathbf{S}_a^{-1} \right)^{-1} \left[ \left( \mathbf{F}_1 + \mathbf{S}_a^{-1} \right) \left( \boldsymbol{\alpha}_1 - \mathbf{A}_1 \mathbf{x}_a \right) + \left( \mathbf{F}_2 + \mathbf{S}_a^{-1} \right) \left( \boldsymbol{\alpha}_2 - \mathbf{A}_2 \mathbf{x}_a \right) \right] =$$
$$= \left( \mathbf{F}_1 + \mathbf{F}_2 + \mathbf{S}_a^{-1} \right)^{-1} \left[ \left( \mathbf{F}_1 + \mathbf{S}_a^{-1} \right) \boldsymbol{\alpha}_1 + \left( \mathbf{F}_2 + \mathbf{S}_a^{-1} \right) \boldsymbol{\alpha}_2 - \mathbf{F}_1 \mathbf{x}_a - \mathbf{F}_2 \mathbf{x}_a \right] = \qquad (12)$$
$$= \left( \mathbf{F}_1 + \mathbf{F}_2 + \mathbf{S}_a^{-1} \right)^{-1} \left[ \left( \mathbf{F}_1 + \mathbf{S}_a^{-1} \right) \boldsymbol{\alpha}_1 + \left( \mathbf{F}_2 + \mathbf{S}_a^{-1} \right) \boldsymbol{\alpha}_2 + \mathbf{S}_a^{-1} \mathbf{x}_a \right] - \mathbf{x}_a$$

where we have used Eq. (9).

Using Eqs (A2) and (A5) of Schneider et al. (2021), we can write the noise error CMs as:

$$\mathbf{S}_{\hat{\mathbf{x}}_i, \mathrm{n}} = \left( \mathbf{F}_i + \mathbf{S}_a^{-1} \right)^{-1} \mathbf{F}_i \left( \mathbf{F}_i + \mathbf{S}_a^{-1} \right)^{-1} \qquad\qquad i = 1, 2 . \qquad (13)$$

In the case that $\mathbf{S}_{\hat{\mathbf{x}}_i, \mathrm{n}}$ are not singular matrices, from Eq. (9) it results:

$$\mathbf{A}_i^T \mathbf{S}_{\hat{\mathbf{x}}_i, \mathrm{n}}^{-1} = \left( \mathbf{F}_i + \mathbf{S}_a^{-1} \right) \qquad\qquad i = 1, 2 \qquad (14)$$

and, using Eqs. (6) and (14), from Eq. (12) we obtain:

$$\hat{\mathbf{x}}_f = \left( \mathbf{A}_1^T \mathbf{S}_{\hat{\mathbf{x}}_1, \mathrm{n}}^{-1} \mathbf{A}_1 + \mathbf{A}_2^T \mathbf{S}_{\hat{\mathbf{x}}_2, \mathrm{n}}^{-1} \mathbf{A}_2 + \mathbf{S}_a^{-1} \right)^{-1} \left[ \mathbf{A}_1^T \mathbf{S}_{\hat{\mathbf{x}}_1, \mathrm{n}}^{-1} \boldsymbol{\alpha}_1 + \mathbf{A}_2^T \mathbf{S}_{\hat{\mathbf{x}}_2, \mathrm{n}}^{-1} \boldsymbol{\alpha}_2 + \mathbf{S}_a^{-1} \mathbf{x}_a \right], \qquad (15)$$

which is the equation of the CDF reported in Eq. (5) of Ceccherini et al. (2015), proving the equivalence of the Kalman filter method reported in Schneider et al. (2021) with the CDF method reported in Ceccherini et al. (2015). The form of Eq. (15) highlights that the combination of the two measurements is a generalization of the weighted mean, where in the weights not

only the CMs, but also the averaging kernel matrices are taken into account. The expression coincides with the weighted mean in the case that the averaging kernel matrices are equal to the identity matrices.

In the case that $\mathbf{S}_{\hat{\mathbf{x}}_i, \mathrm{n}}$ are singular matrices, we cannot write Eq. (15). However, using Eq. (A4) of Schneider et al. (2021) and Eq. (6), from Eq. (12) we obtain:

$$\hat{\mathbf{x}}_f = \left( \mathbf{S}_{\hat{\mathbf{x}}_1}^{-1} \mathbf{A}_1 + \mathbf{S}_{\hat{\mathbf{x}}_2}^{-1} \mathbf{A}_2 + \mathbf{S}_a^{-1} \right)^{-1} \left[ \mathbf{S}_{\hat{\mathbf{x}}_1}^{-1} \boldsymbol{\alpha}_1 + \mathbf{S}_{\hat{\mathbf{x}}_2}^{-1} \boldsymbol{\alpha}_2 + \mathbf{S}_a^{-1} \mathbf{x}_a \right]. \qquad (16)$$

Eq. (16) is valid also when $\mathbf{S}_{\hat{\mathbf{x}}_i, \mathrm{n}}$ are singular matrices and fully maintains the equivalence with the method proposed in

Schneider et al. (2021), furthermore, it is equivalent to Eq. (15) when $\mathbf{S}_{\hat{\mathbf{x}}_i, \mathrm{n}}$ are not singular matrices. Therefore, Eq. (16) is more general than Eq. (15) proposed for the CDF in Ceccherini et al. (2015).

## 3 Conclusions

The equivalence between the method for the synergetic use of data acquired by different instruments proposed in Schneider et al. (2021) and the CDF method proposed in Ceccherini et al. (2015) has been proved. However, the original CDF formula

can only be used in the case that the noise error CMs of the fused products are not singular. The full equivalence of the two methods exists only in the case of the revised CDF formula that has been here derived and given in Eq. (16).

The CDF method was also proved (Ceccherini, 2016) to be equivalent to the measurement-space-solution data fusion method (Ceccherini et al., 2009) that explicitly considered the case of measurements made in an incomplete space. Therefore, the three methods are equivalent among themselves and, for linear and moderately nonlinear problems, they are

90 all equivalent to the simultaneous retrieval. Consequently, we expect that the choice of which among these three methods can be more efficiently used in an operational data fusion depends only on the implementation advantages. Studies that assess the implementation convenience of the three methods for different data fusion problems would be very useful. A significant difference in the implementation between the method proposed in Schneider et al. (2021) and the other two methods occurs when we have to combine more than two measurements. In this case, while the Kalman filter method

requires a sequential approach in which we add one measurement at a time, the CDF and the measurement-space-solution data fusion methods have the advantage that can be implemented fusing all the available measurements in a single step.

*Competing interests.* The author declares that he has no conflict of interest.

*Acknowledgments.* The author is grateful to Bruno Carli for useful discussions.

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
