# Peer review of "Comment on “Synergetic use of IASI and TROPOMI space borne sensors for generating a tropospheric methane profile product”"

_Atmospheric Measurement Techniques, 2021_

## Author Response (AR1)

I thank the reviewers for the useful comments. In the following, I answer the specific comments (included in "**boldface**" for clarity) and, whenever required, I describe the related changes implemented in the revised manuscript. Page and line numbers indicated refer to the original version of the Comment published on AMTD.

**Anonymous Referee #1**

**This Comment on "Synergetic use of IASI and TROPOMI space borne sensors for generating a tropospheric methane profile product by Schneider et al. (2021)" is necessary to compliment the original paper in terms of comparison/reference of existing data fusion methods. If Schneider et al. (2021) had properly referenced Warner et al. (2014) and Ceccherini et al. (2015) with good understanding of the similarities and differences, this Comment would not have been necessary. This Comment clearly and thoroughly compared the Schneider et al. (2021) approach, which was published by Warner et al. (2014) for a different data set, and that of Ceccherini et al. (2015) via derivation of equations. The Comment also reiterated the similarity of the above approaches to that of Ceccherini et al. (2009), which was discussed by Schneider et al. (2021). This reviewer recommends the publication of this Comment.**

The referee recommends the publication of the Comment as it is.

**Anonymous Referee #2**

**The main purpose of this paper was to prove the equivalency of two methods in previous literature, Schneider et al. (2021) and Ceccherini et al. (2015). Overall, this work does a good job in doing this and the mathematical derivation in Section 2 is sound. The manuscript could use additional information that would be useful for the reader - the background and conclusions can be slightly extended to better express why this study is significant. I provide some suggestions below that would be beneficial to implement in the paper. However for the most part, the quality of the pre-print is good, and with a few minor edits I would recommend it for publication.**

I will provide a revised version of the Comment with implemented the suggestions of the referee.

**Specific comments:**

**Introduction: It would be helpful to more clearly state the significance and motivation behind proving the equivalence of the two methods. I understand this is mostly a technical study but it would good for the reader to know the scientific implications of the derivation.**

In the revised version of the Comment, I added a few sentences at the end of the introduction that clarify the significance and the motivation behind proving the equivalence of the two methods.

**Introduction: A few sentences that summarize the retrieval techniques and how they work would be useful to set the background behind this study.**

In the revised version of the Comment, I added a few sentences at the beginning of the introduction that summarize the retrieval techniques and how they work.

**Section 2: Define $F_1$ and $F_2$ and $F_i$ in the equation (Eq 5 and 6). Was not completely clear to me what the relationship between those variables are. Also a sentence describing the physical meaning behind the key equations would be helpful (for example Eq 1, 5 and 15).**

In the revised version of the Comment, I slightly modified the sentence that defines $\mathbf{F}_1$ and $\mathbf{F}_2$ and I added a sentence describing the physical meaning of these matrices. Furthermore, I added sentences at the end of the paragraphs including Eq. (1), Eq. (5, 6) and Eq. (15) describing the physical meaning of these equations.

**Conclusions: The main point of the paper was proving the equivalence of the two methods which shows the similarities between them. Is there any fundamental difference between them? If possible, it would be useful to also mention some advantages of using one method over the other when performing satellite retrievals.**

I have identified a significant difference of implementation in the case that we have to combine more than two measurements. In this case, the CDF has an advantage with respect to the Kalman filter method that I have described at the end of the conclusions in the revised version of the Comment.

**Technical corrections:**

**Line 14 and Line 17: Spell out the names of the sensors where abbreviations are used for the first time. i.e. TROPOMI, IASI, MLS…**

In the revised version of the Comment, I spelled out the names of the sensors.

**Line 60-62: The sentence "However, the original CDF formula…" is a bit long. It can potentially be split into two sentences.**

In the revised version of the Comment, I split the sentence into two sentences.

**Line 64-65: The sentence "Case that corresponds to having singular matrices" was a bit confusing. Consider rewording this part.**

In the revised version of the Comment, I deleted this sentence.

[revised manuscript text omitted]

---

## Referee Report (RR1)

Dear Simone Ceccherini,

Dear Editor,

I am the leading author of the Schneider et al. (2021) manuscript, on which this comment is focusing. I would like to thank Simone Ceccherini for the interest in our work and the Editor for taking into account my thoughts.

The comment of Simone Ceccherini refers exclusively to the theoretical part of our work (the appendix). The text and the equations are in general easy to follow (for dedicated experts with respective background knowledge). In the following, I refer to the line and equation numbers of the LATEXdiff document.

General comments:

I am a bit confused about a comment manuscript focusing exclusively on a paper that is still in revision and that revised work (i.e., already available improvements/clarifications/extensions of the preliminary work) is not considered.

We got detailed comments on the Schneider et al. (2021) manuscript from four different colleagues/referees. The revision of the theoretical part has already been finalized end of March and published as reply to Referee #1 on 29 March (https://doi.org/10.5194/amt-2021-31-AC1). We are currently finalizing revisions, improvements and extensions linked to the comments on using different observational and model data. We hope that the fully revised manuscript can soon be published.

I think that the initiative of Simone Ceccherini for discussing similarities (and differences) of the different data fusion methods can be useful and important. I am also grateful to Simone Ceccherini for providing a helpful referee comment on our paper (https://doi.org/10.5194/amt-2021-31-CC1). However, I am not convinced about the publication of a full manuscript that exclusively comments on work that is still in revision. In my opinion such full manuscript makes sense if it refers to finally revised work (not preliminary work). I know that we are probably taking a lot of time for our final revisions and I would like to apologize for that. However, I can only promise that we do our best in the time we have available.

In case there are constraints that require a very fast publication in AMT of this comment, I strongly suggest commenting on the revised theoretical part of our work, which has already been published in March (as the reply to Referee #1, https://doi.org/10.5194/amt-2021-31-AC1). Due to the discussion with Referee #1, we were able to better clarify when a Kalman filter approach is useful and when not. Furthermore, we put special emphasis on avoiding equations where singular (or potentially singular) matrices have to be inverted and the Simone Ceccherini comment should be adjusted accordingly (e.g. it references in line 52 to an equation that is avoided in our revised work). In the following I list the major modification that I think should be considered before a publication of the Simone Ceccherini comment.

In the revised theoretical part (Appendix A of https://doi.org/10.5194/amt-2021-31-AC1) we distinguish two situations. We start with discussing the combination of two profile retrieval products (see Appendix A2.1). There the application of a Kalman filter is not needed and the combination can be easily achieved

by using the a posteriori covariances of the two individual profile retrievals (see Eq. A11). This completely avoids the problem of a potentially singular retrieval noise matrix. This Eq. (A11) is the same as the second line of Eq. (10) of the Simone Ceccherini comment. So the comment paper could start already with Eq. (10). Then the comment shows the equivalence of Eq. 10 (i.e. our Eq. A11) with Eq. (16), the Complete Data Fusion (CDF) method. However, I do not understand the advantage of Eq. (16), if compared to Eq. (10). Actually Eq. (10) (i.e. our Eq. A11) needs less input than Eq. (16). The first equation needs the retrieval products, the a priori data, the a priori covariances, and the a posteriori covariances. The second equation needs the same input and in addition the two averaging kernels. So, why should one use Eq. (16) instead of Eq. (10)?

In Appendix A2.2 of our revised theoretical part ([https://doi.org/10.5194/amt-2021-31-AC1](https://doi.org/10.5194/amt-2021-31-AC1)) we discuss the combination of a profile and a column data product. This is the problem, on which our work is focusing. It is only this situation, for which we suggest the application of the Kalman filter approach. We show the large similarity of the Kalman filter approach and a combined retrieval that uses the two individual measurements. This combination of column data with profile data is not captured by the CDF (Complete Data Fusion) method as written in Eq. (16) of Simone Ceccherini's comment, because for a column observation, the profile averaging kernels and profile a posteriori covariances are not readily available (if they can be made available at all), but both are needed in the respective Eq. (16).

In general and if data with different vertical representations (fine and coarse) are going to be combined, the CDF method (Eq. 16 of Simone Cecherini's comment) can only work on the coarse vertical representation. The averaging kernel and the a posteriori covariances of the fine gridded profile can be interpolated to the coarse grid (e.g., von Clarmann and Grabowski, [https://doi.org/10.5194/acp-7-397-2007](https://doi.org/10.5194/acp-7-397-2007)), but not vice versa. This means that when combining a profile product with a column product, the CDF method can (to my understanding) only generate a combined column product. In contrast, our Kalman filter-based approach can combine profile and column data and generate a profile observation that has an improved vertical sensitivity.

Specific comments:

Related to the aforementioned last general comment: when discussing pros/cons of the different methods (end of the conclusion section), it should be mentioned, that our Kalman filter method allows for generating a profile observation with improved vertical sensitivity by combining a profile and a total column observation. This is of large importance for greenhouse gas research, where high precision total column observations are frequently used. To my understanding and according to Eq. 16 of Simone Ceccherini's comment such profile and column data combination is not possible by the CDF method.

Line 27 – 29 (level of equivalency of Warner et al. 2014 and our work): Warner et al. (2014) had the same principle idea as we, i.e. the application of a Kalman filter for combining different satellite sensor observations. However, there are also important differences. (1) Warner et al. (2014) uses horizontal fields measured by AIRS with weak vertical details as the background and focuses on improving the vertical information for this large area by using the detailed vertical information provided very locally by the observations of TES (and MLS). In their method the vertical information comes mainly from TES (or MLS, see their Figs. 6, 7 and 11) and the horizontal information from AIRS. In our method we use two sensors, which both have good horizontal coverage (we do not need an analysis in the horizontal

dimension), but different and rather synergetic vertical sensitivities. In our method we optimally combine the different vertical sensitivities and generate a combined observation that has more detailed vertical information than each of the two individual observations. (2) In Warner et al. (2014) the combination is not made in a fully optimal sense (that would be equivalent to the combined optimal retrieval products). They use a diagonal observational error covariance matrix (**R** in their Eq. 2) and a global statistics for the satellite sensors' noise errors and sensitivities (see their Figs 2 and 3 and the related text). In contrast, our method works with the individual noise errors and sensitivities of exactly the two observations that are combined, i.e. our method is rather similar to a combined optimal estimation retrieval product.

Equation (14): I was not able to get from Eq. (13) to Eq. (14). One has to write out the inverse of Eq. (13), right? However, this means that there is the term $\mathbf{F}_i^{-1}$, which however, does not exist. The impossibility of calculating $\mathbf{F}_i^{-1}$ is actually the reason for imposing the side constraint ($\mathbf{S}_a^{-1}$). Or is there another way to get from Eq. (13) to Eq. (14) that avoids the use of $\mathbf{F}_i^{-1}$?

Best regards,

Matthias Schneider

---

## Author Response (AR2)

I thank the reviewers for the comments. In the following, I answer the specific comments (included in "**boldface**" for clarity) and, whenever required, I describe the related changes implemented in the revised manuscript.

**Referee #1, Juying Warner**

The referee recommends the publication of the Comment as it is.

**Referee #3, Matthias Schneider**

**Dear Simone Ceccherini,**
**Dear Editor,**
**I am the leading author of the Schneider et al. (2021) manuscript, on which this comment is focusing. I would like to thank Simone Ceccherini for the interest in our work and the Editor for taking into account my thoughts.**
**The comment of Simone Ceccherini refers exclusively to the theoretical part of our work (the appendix). The text and the equations are in general easy to follow (for dedicated experts with respective background knowledge). In the following, I refer to the line and equation numbers of the LATEXdiff document.**

**General comments:**
**I am a bit confused about a comment manuscript focusing exclusively on a paper that is still in revision and that revised work (i.e., already available improvements/clarifications/extensions of the preliminary work) is not considered.**
**We got detailed comments on the Schneider et al. (2021) manuscript from four different colleagues/referees. The revision of the theoretical part has already been finalized end of March and published as reply to Referee #1 on 29 March (https://doi.org/10.5194/amt-2021-31-AC1). We are currently finalizing revisions, improvements and extensions linked to the comments on using different observational and model data. We hope that the fully revised manuscript can soon be published.**
**I think that the initiative of Simone Ceccherini for discussing similarities (and differences) of the different data fusion methods can be useful and important. I am also grateful to Simone Ceccherini for providing a helpful referee comment on our paper (https://doi.org/10.5194/amt-2021-31-CC1). However, I am not convinced about the publication of a full manuscript that exclusively comments on work that is still in revision. In my opinion such full manuscript makes sense if it refers to finally revised work (not preliminary work). I know that we are probably taking a lot of time for our final revisions and I would like to apologize for that. However, I can only promise that we do our best in the time we have available.**
**In case there are constraints that require a very fast publication in AMT of this comment, I strongly suggest commenting on the revised theoretical part of our work, which has already been published in March (as the reply to Referee #1, https://doi.org/10.5194/amt-2021-31-AC1). Due to the discussion with Referee #1, we were able to better clarify when a Kalman filter approach is useful and when not. Furthermore, we put special emphasis on avoiding equations where singular (or potentially singular) matrices have to be inverted and the Simone Ceccherini comment should be adjusted accordingly (e.g. it references in line 52 to an equation that is avoided in our revised work). In the following I list the major modification that I think should be considered before a publication of the Simone Ceccherini comment.**

The purpose of the Comment is to show the equivalence between the Kalman filter method and the Complete Data Fusion (CDF) method and I think that the very good description of the Kalman filter method given by the authors in the version published on AMTD is much useful to show this equivalence. The revised theoretical part is not equally good for this purpose. Furthermore, it is possible that the revised part will be subjected to review, therefore, it is possible that neither this will be that published on AMT. Therefore, I prefer to maintain the reference to the equations given in the AMTD version, as it is in the intent of published discussion papers. In the Comment the presence of matrices potentially singular in the formulas is analyzed and discussed. The equation in line 52 does not have singularity problems.

**In the revised theoretical part (Appendix A of https://doi.org/10.5194/amt-2021-31-AC1) we distinguish two situations. We start with discussing the combination of two profile retrieval products (see Appendix A2.1). There the application of a Kalman filter is not needed and the combination can be easily achieved by using the a posteriori covariances of the two individual profile retrievals (see Eq. A11). This completely avoids the problem of a potentially singular retrieval noise matrix. This Eq. (A11) is the same as the second line of Eq. (10) of the Simone Ceccherini comment. So the comment paper could start already with Eq. (10). Then the comment shows the equivalence of Eq. 10 (i.e. our Eq. A11) with Eq. (16), the Complete Data Fusion (CDF) method. However, I do not understand the advantage of Eq. (16), if compared to Eq. (10). Actually Eq. (10) (i.e. our Eq. A11) needs less input than Eq. (16). The first equation needs the retrieval products, the a priori data, the a priori covariances, and the a posteriori covariances. The second equation needs the same input and in addition the two averaging kernels. So, why should one use Eq. (16) instead of Eq. (10)?**

It is not true that Eq. (16) needs more inputs than Eq. (10). If the equations are equivalent the same information is used and the relative convenience does not depend on the number of terms, but on the terms that are available for the input data. However, even the different number of independent matrices (the additional two averaging kernels of the second equation) is only an apparent difference due to the fact that in the calculations, there is the assumption, made by the authors of the paper, that the same a priori covariance matrix is used both in the retrieval of the individual profiles and in the formula of the fusion. In general, the $\mathbf{S}_a$s of Eq. (10) are not the same: the one added to $\mathbf{F}_1$ is used in the retrieval of $\hat{\mathbf{x}}_1$, the one added to $\mathbf{F}_2$ is used in the retrieval of $\hat{\mathbf{x}}_2$ and the one used in the first term corresponds to the a priori covariance matrix chosen for the fusion. In order to express the $\mathbf{F}_1$ and $\mathbf{F}_2$ matrices of the first term of Eq. (10) as a function of the $\mathbf{S}_{\hat{\mathbf{x}}_1}$ and $\mathbf{S}_{\hat{\mathbf{x}}_2}$ matrices it is necessary to know the a priori covariance matrices used in the retrievals of $\hat{\mathbf{x}}_1$ and $\hat{\mathbf{x}}_2$. Therefore, in order to use Eq. (10) it is necessary to know the two a priori covariance matrices used in the retrievals of $\hat{\mathbf{x}}_1$ and $\hat{\mathbf{x}}_2$ and to decide which a priori covariance matrix to use for the fused product. Instead, in Eq. (16) only the a priori covariance matrix used in the fusion explicitly appears while the information about the a priori covariance matrices used in the retrievals of $\hat{\mathbf{x}}_1$ and $\hat{\mathbf{x}}_2$ is contained in the respective averaging kernel and covariance matrices, causing the additional two averaging kernels in the second equation. These "additional" averaging kernels are in place of the missing a priori covariance matrices of the fusing products. Therefore, the two equations have also the same number of independent terms.

**In Appendix A2.2 of our revised theoretical part (https://doi.org/10.5194/amt-2021-31-AC1) we discuss the combination of a profile and a column data product. This is the problem, on which our work is focusing. It is only this situation, for which we suggest the application of the Kalman filter approach. We show the large similarity of the Kalman filter approach and a combined retrieval that uses the two individual measurements. This combination of column data with profile data is not captured by the CDF (Complete Data Fusion) method as written in Eq. (16) of Simone Ceccherini's comment, because for a column observation, the profile averaging**

**kernels and profile a posteriori covariances are not readily available (if they can be made available at all), but both are needed in the respective Eq. (16).**

The description of how to use the CDF to fuse a profile and a column is provided in:
C. Tirelli, S. Ceccherini, N. Zoppetti, S. Del Bianco, M. Gai, F. Barbara, U. Cortesi, J. Kujanpää, Y. Huan and R. Dragani, *Data Fusion Analysis of Sentinel-4 and Sentinel-5 Simulated Ozone Data*, Journal of Atmospheric and Oceanic Technology, Vol. **37**, No. 4, 573 (2020), doi: 10.1175/JTECH-D-19-0063.1.
In the paper, the procedure is also applied to simulated measurements.

**In general and if data with different vertical representations (fine and coarse) are going to be combined, the CDF method (Eq. 16 of Simone Cecherini's comment) can only work on the coarse vertical representation. The averaging kernel and the a posteriori covariances of the fine gridded profile can be interpolated to the coarse grid (e.g., von Clarmann and Grabowski, https://doi.org/10.5194/acp-7-397-2007), but not vice versa. This means that when combining a profile product with a column product, the CDF method can (to my understanding) only generate a combined column product. In contrast, our Kalman filter-based approach can combine profile and column data and generate a profile observation that has an improved vertical sensitivity.**

The use of the CDF to fuse profiles represented on different vertical grids is described in:
S. Ceccherini, B. Carli, C. Tirelli, N. Zoppetti, S. Del Bianco, U. Cortesi, J. Kujanpää, and R. Dragani, *Importance of interpolation and coincidence errors in data fusion*, Atmospheric Measurement Techniques, **11**, 1009–1017 (2018), doi: 10.5194/amt-11-1009-2018
The CDF can work both with coarser and finer grids with respect to those of the individual retrieved profiles. Furthermore, the approach described in Tirelli et al. (2020) to fuse a column and a profile generates as output a profile with improved vertical sensitivity.

**Specific comments:**

**Related to the aforementioned last general comment: when discussing pros/cons of the different methods (end of the conclusion section), it should be mentioned, that our Kalman filter method allows for generating a profile observation with improved vertical sensitivity by combining a profile and a total column observation. This is of large importance for greenhouse gas research, where high precision total column observations are frequently used. To my understanding and according to Eq. 16 of Simone Ceccherini's comment such profile and column data combination is not possible by the CDF method.**

As already mentioned above the profile and column data combination can be done also with the CDF method, as described in Tirelli et al. (2020).

**Line 27 – 29 (level of equivalency of Warner et al. 2014 and our work): Warner et al. (2014) had the same principle idea as we, i.e. the application of a Kalman filter for combining different satellite sensor observations. However, there are also important differences. (1) Warner et al. (2014) uses horizontal fields measured by AIRS with weak vertical details as the background and focuses on improving the vertical information for this large area by using the detailed vertical information provided very locally by the observations of TES (and MLS). In their method the vertical information comes mainly from TES (or MLS, see their Figs. 6, 7 and 11) and the horizontal information from AIRS. In our method we use two sensors, which both have good horizontal coverage (we do not need an analysis in the horizontal dimension), but different**

**and rather synergetic vertical sensitivities. In our method we optimally combine the different vertical sensitivities and generate a combined observation that has more detailed vertical information than each of the two individual observations. (2) In Warner et al. (2014) the combination is not made in a fully optimal sense (that would be equivalent to the combined optimal retrieval products). They use a diagonal observational error covariance matrix (R in their Eq. 2) and a global statistics for the satellite sensors' noise errors and sensitivities (see their Figs 2 and 3 and the related text). In contrast, our method works with the individual noise errors and sensitivities of exactly the two observations that are combined, i.e. our method is rather similar to a combined optimal estimation retrieval product.**

In the revised version of the manuscript, I modified the sentence writing that the method of Warner et al., (2014), is based on the same principle of that presented in the paper of Schneider et al. (2021), i.e. the application of the Kalman filter for combining different satellite sensor observations.

**Equation (14): I was not able to get from Eq. (13) to Eq. (14). One has to write out the inverse of Eq. (13), right? However, this means that there is the term F$i$-1, which however, does not exist. The impossibility of calculating F$i$-1 is actually the reason for imposing the side constraint (S$a$-1). Or is there another way to get from Eq. (13) to Eq. (14) that avoids the use of F$i$-1?**

Just before Eq. (14) there is the hypothesis that $\mathbf{S}_{\hat{x}_i,n}$ are not singular matrices and, as a consequence, the matrices $\mathbf{F}_i$ are not singular. Therefore Eq. (14) is obtained from Eq. (13) in this hypothesis. The case in which $\mathbf{S}_{\hat{x}_i,n}$ are singular matrices is discussed below and it is the reason why Eq. (16) is introduced.